# Validation of a New High-Throughput Cell Separation Method for Downstream Molecular Applications

**DOI:** 10.3390/ijms26146747

**Published:** 2025-07-14

**Authors:** Daisy Shillingford, Andreas Radek, Andrea Kupitz, Rebecca Thomas, Christopher Connor, Stuart Paul Adams

**Affiliations:** 1Great Ormond Street Hospital for Children NHS Foundation Trust, London WC1N 3BH, UK; daisy.shillingford@gosh.nhs.uk (D.S.); rebecca.thomas@gosh.nhs.uk (R.T.); chrisopher.connor@gosh.nhs.uk (C.C.); 2Miltenyi Biotec, 51429 Bergisch Gladbach, Germany; andreasra@miltenyi.com (A.R.);

**Keywords:** cell sorting, T cells, B cells, granulocytes, chimerism

## Abstract

The development of new cell separation technologies has continued as the demand for sorted cell populations for molecular testing increases. The goal is to increase through-put potential and reduce the manual handling of samples required whilst ensuring that cells are sorted efficiently with high purity. Herein, we review two affinity-based methods utilising magnetic beads to isolate cells: one is currently used within a clinical laboratory as standard of care and the other is a newly developed larger platform using the same principle. Cells were sorted simultaneously on both platforms and assessments were made of the purity, cell recovery, and hands-on time, indicating that the new larger platform is sufficient for use in a clinical laboratory as it not only increased cell sorting capacity and reduced manual processing but was also able to isolate cells with sufficient purity levels for downstream molecular testing.

## 1. Introduction

Clinical laboratories across the world routinely sort cell populations from blood and bone marrow samples for downstream molecular testing. Cell sorting technologies are also utilised in research laboratories to isolate various cell types for lineage-specific analysis. Many technologies exist, but one of the most common approaches is affinity-based cell separation [1], which uses antibodies conjugated to nano-sized magnetic beads targeting cluster of differentiation (CD) antigens for positive selection of the target cell. Once the antibody has bound to the CD antigen, the sample is passed through a column with a high-gradient magnetic field, thus allowing the bound cells to be separated due to the magnetic force on the bead [2,3]; this process is referred to as magnetic-activated cell sorting (MACS).

Until recently, MACS technologies were entirely manual, an example being the MiniMACS supplied by Miltenyi Biotec, which went on to develop the partially automated cell separator, the autoMACS Pro Separator (AMP), which enabled laboratories to process more samples with less direct sample handling involvement. This is now commonly used in routine clinical laboratories as demand for lineage-specific testing has increased, as has the requirement for automation. The continual rise in the use of haematopoietic stem cell transplants (HSCT) to successfully treat a range of conditions [4,5] has resulted in the need for higher throughput cell separation technology due to the increased sensitivity achieved by testing lineage-specific cells over whole blood [6,7]. Post-HSCT patients will have their chimerism (donor engraftment) levels monitored as standard of care at set timepoints as outlined by the European Society for Blood and Bone Marrow [6]. After this, the patients are likely to be monitored for at least a year, with many being tracked for the remainder of their lives to identify late graft rejection [8]. Therefore, sample numbers are set to increase each year. To provide an accurate assessment of donor engraftment levels, the purity of the sorted cells is imperative, as studies have shown that variable levels of engraftment may be detected in each cell lineage post-HSCT, with significant clinical implications depending on the underlying condition [5,8]. Consequently, contamination of the separated cells due to poor cell sorting can impact the patient’s results, skewing them due to the presence of other cell types; thus, the purity of all separated cells is assessed prior to reporting specific cell lineage chimerism results [9].

In addition, cell separations are performed for many other downstream clinical tests. In our laboratory, these include T cell evaluation, such as the measurement of T cell receptor excision circles (TRECs), which acts as a reflection of thymic output [10] and is performed on extracted DNA, and Vβ Spectratyping, which provides an overview of the patient’s T cell repertoire via the extraction of RNA from the separated cells and identifying chronic Epstein–Barr viral infections that may be present in not only B cells but CD3+ T cells and CD56+ NK cells [11,12]. These tests are clinically significant as they provide diagnostic and prognostic information for patients with a variety of immune conditions [9,10,13]. For TRECs and Spectratyping assays, sufficient cell recovery is crucial to accurately assess patients; therefore, both assays include the evaluation of a control gene to ensure that an adequate number of cells have been separated.

Other cell sorting techniques available include RosetteSep™, which isolates cells via negative selection, resulting in the removal of non-target cells [14]. This method offers both good cell recovery and purity as the target cells remain in solution and completely unaltered for downstream processing [1]. However, as this method requires the non-target cells to bind to the red blood cells present in the patients’ sample, the hematocrit level can affect the cell separation process [14]. Moreover, this technique has multiple manual steps, which would not support a high-throughput laboratory. Dynabeads are also available for cell separation and operate using magnetic beads to positively select target cells from a whole-blood sample [15]. They also isolate cells with a high purity and yield, and there have been developments to use them in point-of-care testing in the form of a microfluidic platform to assess CD4+ T cell counts [16]. Similarly, to the RosetteSep™ method, Dynabeads require significant manual handling compared to the autoMACS Pro, and a review of a range of Dynabeads that are available has demonstrated variability in the products in terms of their applicability to the downstream processing of the separated cells [17]. The manual handling involved is counterproductive as the current demand on clinical laboratories performing molecular testing rises; therefore, this has led to the further generation of a new fully automated, increased throughput cell separator by Miltenyi Biotec, the MultiMACS X Separator (MMX), which is evaluated herein. We demonstrate herein the first use of a high-throughput system in a clinical setting on human whole-blood samples for downstream molecular testing, such as cell lineage chimerism analysis post-transplant, establishing the benefits of time saving, reduction in handling errors, and sustained good quality of both sample purity and yield.

## 2. Results

Key metrics to determine the performance of the MMX include (1) the purity of the sorted cells, (2) cell recovery, and (3) the time required per sample for sorting. The purity assessment was performed on viable cells only due to the required downstream testing.

### 2.1. Purity of Sorted Cells

The purity of the sorted CD3+ T cells, CD15+ granulocytes, and CD20+ B cells was assessed using flow cytometry. Example flow plots are shown in Figure 1. CD20+ was required for B cell purity assessment due to the CD19+ epitope being blocked by the bound MicroBead, as advised by the manufacturer. The total purity data generated from the cells separated on the MMX included 20 samples from CD3+ T cells with a median purity of 97.5%, CD15+ granulocytes with a median purity of 99.5%, and CD19+ B cells with a median purity of 88.5%. An assessment of cell viability showed that cells separated on the MMX had a median cell viability of 81% for CD3+ T cells, 83% for CD15+ granulocytes, and 75% for CD19+ B cells, indicating improved overall cell viability compared to the current standard-of-care approach, which indicated median cell viability of 70% for CD3+ T cells, 73% for granulocytes, and 77% for CD19+ B cells.

### 2.2. Comparison of Sorted Cells Using Standard of Care and the MMX

The full head-to-head comparison of T cell, B cell, and granulocyte purities between standard-of-care and MMX sorting is shown in Table 1. The same sample was used for both sorting methodologies, and sorting was performed on the same day using the same batch of MACSprep Chimerism MicroBeads.

The purity of the sorted CD3+ T cells was equal to or higher than standard-of-care processing in 14/20 (70%) samples separated with the MMX. Only one specimen (sample 9) had a noticeably reduced purity with the MMX when compared to standard of care. For this sample, there was a delay of 24 h between SoC purity assessment and MMX purity assessment for the T and B cells due to workload commitments. This may have been the cause of the discrepancy. In total, 17/20 MMX sorted T cell samples had a purity >85% (the clinical laboratory’s minimum acceptable purity for downstream reporting). The purity of the sorted CD15+ granulocytes was equal to or higher than standard-of-care processing, in 14/20 (70%) samples separated with the MMX. All samples sorted for CD15+ granulocytes had a purity >85%. The purity of the sorted CD19+ B cells was equal to or higher than standard-of-care processing, in 14/20 (70%) samples separated with the MMX. The majority of samples (70%) sorted for CD19+ B cells had a purity >85%. Plots showing standard of care purities vs MMX purities for each cell type are shown in Figure 2.

### 2.3. Statistical Analysis of Purity Data

The Shapiro–Wilk test for normality demonstrated that none of the T cell, B cell, or granulocyte purity datasets had a normal distribution (*p* < 0.060, *p* < 0.037, and *p* < 0.027, respectively). Thus, the non-parametric Mann–Whitney U test was carried out to establish any significant differences between the purities of cells sorted on the AMP and the MMX. For sorted CD3+ T cells, there was no significant difference between the two methods compared (*U* = 187, *p* = 0.723). For sorted CD15+ granulocytes, there was no significant difference between the two methods compared (*U* = 169.5, *p* = 0.380). For sorted CD19+ B cells, there was no significant difference between the two methods compared (*U* = 200, *p* = 1.000).

The median T cell purity was 97.5% (interquartile range 88.5%, 99%), the median granulocyte purity was 99% (99%, 100%), and the median B cell purity was 89% (77%, 98%). This indicates that the greatest statistical dispersion was observed in the sorted B cells.

### 2.4. Cell Recovery

To determine cell recovery, we obtained absolute T cell, B cell, and granulocyte counts in the primary specimens prior to cell sorting. We then measured the cell count after cell sorting and calculated the total target cell recovery. The results are shown in Table 2.

Most specimens had >50% cell recovery regardless of the cell type targeted, providing enough material for all downstream molecular analyses.

### 2.5. Time Required per Sample for Sorting

Our current standard-of-care processing permits for 5 cell sorts to be completed in a single batch. The MultiMACS X (Miltenyi Biotech B.V. & Co. KG, Bergisch Gladbach, Germany) allows for 24 cell sorts to be carried out simultaneously. Figure 3 provides an overview of the procedures, with the time needed for each step. Figure 4 shows the hands-on time and processes for the standard of care method currently employed.

Overall, the time calculated per cell sort using the MMX was 7 min. This is derived from an assumption that 24 samples are being sorted in each run, which includes all specimen preparatory steps and maintenance processes for the machine. This compares to 20 min with our current standard-of-care approach. Actual scientist hands-on time was more similar between the two methods, with the MMX requiring slightly less than standard-of-care fractionations.

## 3. Discussion

We developed a bespoke high-throughput process for magnetic bead automated cell sorting on an existing commercial platform, primarily for downstream molecular chimerism analysis, but also for other molecular applications. The ultimate aims were to be able to increase workflow, reduce hands-on time, and reduce the risk of user error when sorting large numbers of patient specimens.

To do this, we developed a strategy that permitted a single program on the commercial platform to be universally suitable for sorting T cells, B cells, and granulocytes from a minimal volume of peripheral blood samples. Despite this single “one size fits all” approach, there is scope for additional programs to be developed on the platform should users require them. To ensure that the newly developed program worked sufficiently well, we tested real-world patient samples directly alongside our currently used standard-of-care approach. The current methodology we employ is already widely used by large numbers of other clinical laboratories and is very similar to other methods utilised by the remaining clinical laboratories that perform downstream chimerism testing [18].

We demonstrate that this new, 24-specimen, automated cell sorting system can perform cell sorting in a much-reduced timeframe without any negative effect on either the purity of the cell fractions or on overall cell recovery, consistent with the currently employed SoC method [19]. In addition, we provide details on both the hands-on time-saving and the total time saved when processing a high number of patient specimens. Through the use of the attached barcode reader, we also conclude that the risk of user error when sorting large numbers of specimens is additionally reduced. In the real-world clinical laboratory scenario, there is an increasing need to be able to cope with high-throughput workloads and an increasing need to be able to provide molecular test results on sorted cell populations [5,19]. We have demonstrated that a platform to accommodate these needs does exist, and we have developed and tested the bespoke program that facilitates this high-throughput cell sorting. Although cell viability is not vital for downstream molecular testing (such as chimerism analysis), it is well established that the use of MicroBeads to isolate different cell types has good cell recovery [19] and little impact on the recovered cells’ structure, function, and activity [20]. Thus, they may be suited to further testing for research purposes.

A recommendation that we would have for future development would be the addition of an automated integrated purity assessment. As cell fraction purity is crucial in downstream testing (e.g., chimerism analysis), this would further reduce the hands-on time required for sample processing and improve sample processing efficiency whilst also reducing the potential for human error. It may also be beneficial to have liquid level sensors included in uptake pipettes, as this would avoid the need for users to manually check that MicroBead volumes are adequate prior to starting.

## 4. Materials and Methods

### 4.1. MultiMACS X Cell Fractionation

Peripheral blood was collected into EDTA tubes from patients requiring cell separation for downstream molecular testing. The 20 samples included in Table 1 are from 8 healthy HSCT donors (used for downstream chimerism short tandem repeat analysis), 4 patients pre-HSCT for chimerism analysis, 1 patient post-thymus transplant for chimerism analysis, and 7 patients post-HSCT for various conditions (leukaemia and metabolic and primary immunodeficiencies) used for subsequent chimerism analysis. The isolated cells were determined based on the patient’s clinical condition for downstream chimerism testing [5]. The MMX (Miltenyi Biotech B.V. & Co. KG, Bergisch Gladbach, Germany) was used to separate cells from whole blood using a custom-made programme that required a starting volume of 2.7 mL for CD3+ T cell and CD19+ B cell sorting and 0.9 mL for CD15+ granulocyte fractionation, with an image of the instrument shown in Figure 5. These volumes were chosen based on the specimen size received (7 mL blood draw) and likely increased prevalence of granulocytes in the specimen compared to lymphocytes. No pre-processing of the primary EDTA blood tube was carried out. Patient EDTA blood was loaded directly into the MMX.

Firstly, all samples were checked for clots as this can interfere with downstream testing, and then an aliquot of 200 µL was taken and a full blood count was performed using the Sysmex XN-2000 (Sysmex, Norderstedt, Germany). The starting white blood cell (WBC) count, lymphocyte count, and granulocyte count (including neutrophils and eosinophils) were recorded alongside the actual sample volume used for cell fractionation. Where the sample volume was insufficient to meet the required volumes, it was topped up using MACS fractionation buffer and mixed via inversion. For both the CD3+ T cell and CD19+ B cell fractionations, the starting counts were obtained using flow cytometric assessment of the patient’s lymphocyte subsets [21].

The loading of the MMX was carried out according to the manufacturer’s instructions, which were provided in a step-by-step fashion via the instrument itself. Once the samples, reagents (MicroBeads, MACS fractionation buffer, Whole Blood Elution buffer), and consumables (single use Multi 24-column block, deep well plates (DWPs), 1000 µL filter tips, 5000 µL filter tips and elution tubes) were onboard, processing was fully automated and took approximately 72 min for 24 cell fractionations. All MicroBead volumes and incubation times were programmed to follow the manufacturer’s protocol. The samples and reagents were loaded into chill racks that were stored in a refrigerator (2–8 °C) for a minimum of 2 h prior to use.

All cell sorting was carried out using the MACSprep™ Chimerism MicroBeads (Miltenyi Biotech B.V. & Co. KG, Bergisch Gladbach, Germany) for their corresponding target. The CD3 MicroBeads (catalogue number 130-050-101) specifically bind to the CD3 antigen present on T cells via a conjugated monoclonal anti-human CD3 antibody (mouse IgG2a). The CD15 MicroBeads (130-111-548) bind to the 3-fucosyl-N-acetyllactosamine (3-FAL) carbohydrate structure present on neutrophils and eosinophils via a conjugated monoclonal anti-human CD15 antibody (mouse IgM). The CD19 MicroBeads (130-111-547) bind to the CD19 antigen present on B cells via a conjugated monoclonal anti-human CD19 antibody (mouse IgG1).

An additional safety feature was incorporated into the instrument software that required all samples to be scanned in via a barcode; the patient’s corresponding elution tube was then scanned to ensure that the barcode details match, indicating that the samples had been loaded correctly. Upon completion, the cells were automatically eluted in 1 mL of Whole Blood Elution Buffer.

### 4.2. Standard-of-Care Cell Fractionation

Standard-of-care cell fractionation was performed using the same batches of MACSprep Chimerism MicroBeads as used for MMX processing. Using the autoMACS Pro requires the MicroBeads to be manually added to the patient’s sample. These were directly added at a bead-to-cell ratio of 50 µL per 1 mL of EDTA blood, and the samples were then mixed and incubated in the refrigerator (2–8 °C) for 15 min. No additional blocking reagent or red blood cell lysis steps were included in either the MMX or SoC methods. The same primary EDTA blood samples were also used, with the same sample volume being used for both methods, and cell sorting was carried out on the same day using the autoMACS Pro and the MMX. The POSSELWB programme was used on the autoMACS Pro to complete cell separation.

### 4.3. Post Cell Fractionation Recovery and Purity Assessments

A 200 µL aliquot of sorted cells was taken for samples requiring purity assessment via flow cytometric methods. The remaining sample was spun at 616 g-force for 5 min, and 500 µL of supernatant was then removed. Post cell counts were performed using the Sysmex XN-2000 by adding 200 µL of Sodium Chloride (NaCl) to resuspend the centrifuged cells and then taking a 200 µL aliquot from this cell suspension. The WBC count, lymphocyte count, and granulocyte count (including neutrophils and eosinophils) were subsequently recorded and used to calculate cell recovery. Cell recovery was calculated by obtaining the starting cell count:(CD3+ T cell count (cells/µL) or granulocyte count (cells/µL) or CD19+ B cell count (cells/µL)) × starting volume (mL) = total starting cell count

Following cell separation, a post count was performed to obtain the number of white blood cells recovered:Post WBC (cells/µL) × elution volume (0.5 mL) = post cell separation count

To assess the number of cells recovered, the following calculation was used:Post cell separation count/total starting cell count × 100 = cell recovery (%)

Post cell fractionation purity assessment was carried out using flow cytometry on the FACS Canto II platform using BD FACSDiva version 9 and FlowJo software version 11 (Becton, Dickinson and Company, Franklin Lakes, NJ, USA). Panels for each of the different cell fractionations were used to assess the purity of the sorted cells, with a minimum of 20,000 events being acquired and analysed. Viable WBCs were gated using CD45+ and 7AAD, with doublets being excluded from the analysis. The CD45-negative gate represents red blood cells, platelets, and debris, which are cells without DNA, therefore not impacting downstream molecular testing. Depending on the type of cell fractionation performed, additional antibodies were used to calculate the percentage of contaminating cells present in the sample. For the CD3+ T cell sort purity assessment, the contaminating CD33+ monocytes were gated, the T cells were assessed by reviewing CD3+/CD19- binding, and then the T cell subpopulations were checked using CD4/CD8. For CD15+ granulocyte sorts, the initial viability gating remained the same, but the antibodies used to assess the purity included CD15, followed by assessment of the CD15-negative cells using CD3 to determine T cell contamination and CD20 to establish B cell contamination. For the B cell sort purity assessment, the CD20-positive cells were measured, and the CD20-negative cells were gated to establish their composition using CD33 (monocytes), CD3 (T cells), and CD15 (granulocytes).

### 4.4. Statistical Analyses

Data were plotted and assessed using the Shapiro–Wilk test for normality. Subsequently, the nonparametric Mann–Whitney U Test was utilised to calculate significant differences between the datasets. All statistical analyses were performed using SPSS Statistics software version 22 (IBM, Armonk, NY, USA). A cut-off value of *p* < 0.05 for statistical significance was used for all analyses.

## 5. Conclusions

Overall, the MultiMACS X has demonstrated that it is able to perform cell separations on large numbers of samples with purities that show no statistical difference when compared to purities achieved using the current SoC approach. Cell recovery did not impact the downstream molecular testing being performed (such as chimerism testing), and the hands-on time for cell separation was reduced when using the MMX. Further data on cell viability may be of interest if the isolated cells are to be used for additional research; however, for molecular testing, both the cells recovered and the purity of the sorted cells were sufficient. The integration of the sample scanning system, reduction in manual handling of samples, and larger cell sorting platform (24 samples) has provided a high-throughput cell sorting platform that can be adapted to suit laboratory requirements through bespoke design, thus helping laboratories cope with growing demand.

## Figures and Tables

**Figure 1 ijms-26-06747-f001:**
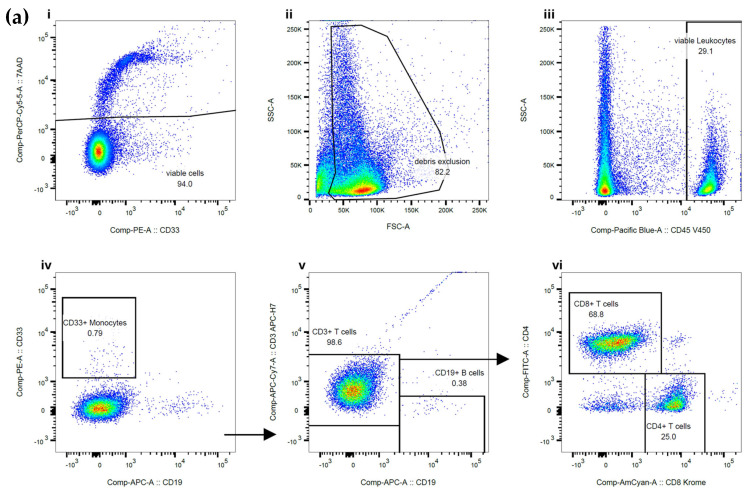
(**a**–**c**). Example plots showing sorted cell purity assessment. Viable WBCs were gated using CD45+ and 7AAD, with debris and doublets being excluded from the analysis. The figures show sequential gating strategies from i–vi. (**a**) For T cell purity, the contaminating CD33+ monocytes were gated, and the total T cell purity was established by reviewing CD3/CD19 binding and checking the T cell subpopulations with CD4/CD8. CD3+ T cell purity in this sample was 98.8%. (**b**) For granulocyte purity, CD15-negative cells were gated to establish their composition using CD33+ (monocytes), CD3 (T cells), and CD20 (B cells). The monocyte and lymphocyte gates shown in (v) are derived from the CD15-negative gate in (iv). Overall, CD15+ granulocyte purity in this sample was 96.8%. (**c**) CD20-negative cells were gated to establish their composition using CD33 (monocytes), CD3 (T cells), and CD15 (granulocytes). Plot (v) gates are derived from the non-B cell population in (iv). Plot (vi) gates are derived from the non-monocyte/non-B cell gate in plot (v). Overall, the B cell purity in this sample was 88.8%.

**Figure 2 ijms-26-06747-f002:**
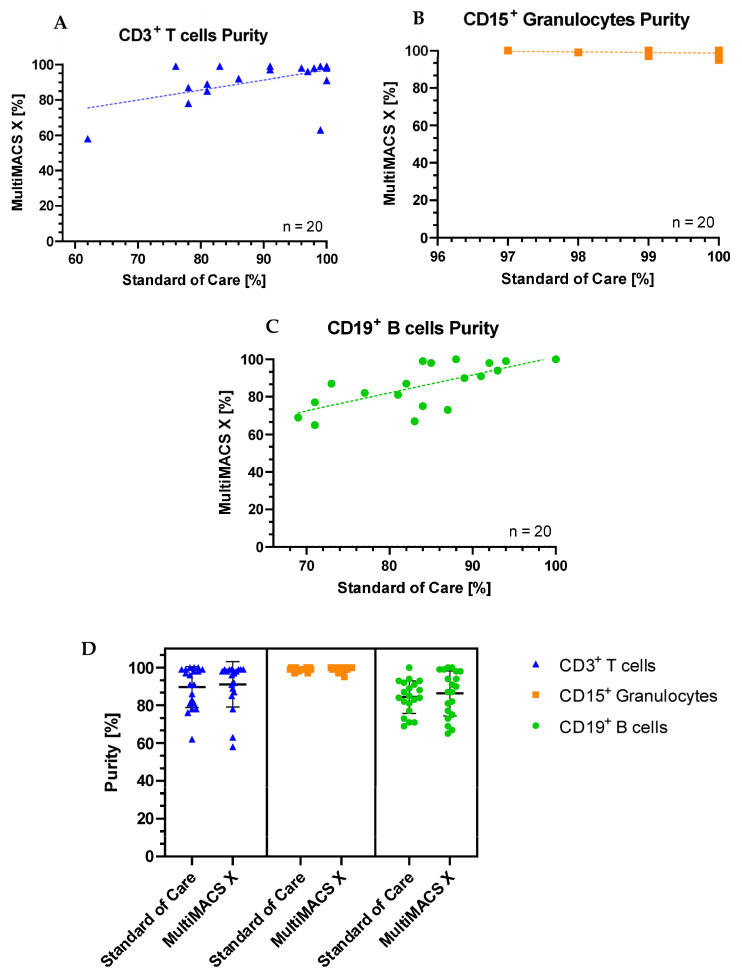
Graphs showing the correlation between standard-of-care sorting and MMX sorting purities for (**A**) CD3+ T cells, (**B**) CD15+ granulocytes, and (**C**) CD19+ B cells. (**D**) shows overall purities by cell fraction with error bars.

**Figure 3 ijms-26-06747-f003:**
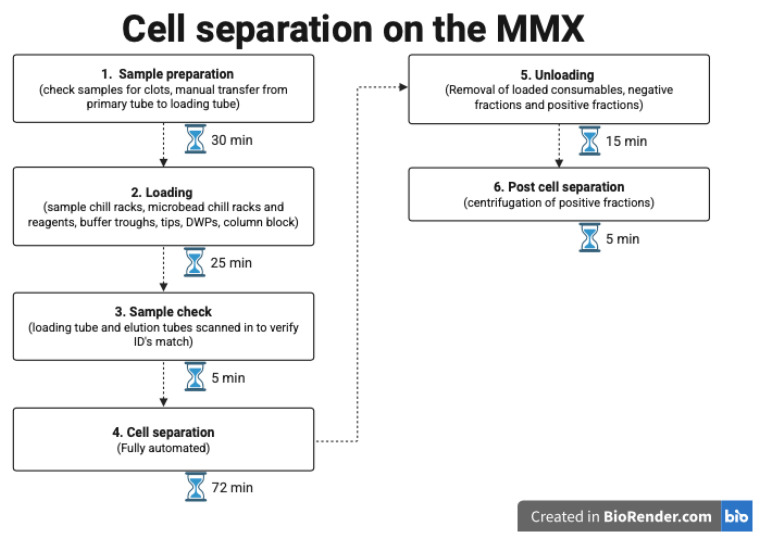
Flowchart showing the steps involved when loading the MMX, including the time required for each step when performing cell separations on 24 samples (created in BioRender version 201). Step 2: Loading deep well plates (DWPs) and Step 3: Sample check identification (ID).

**Figure 4 ijms-26-06747-f004:**
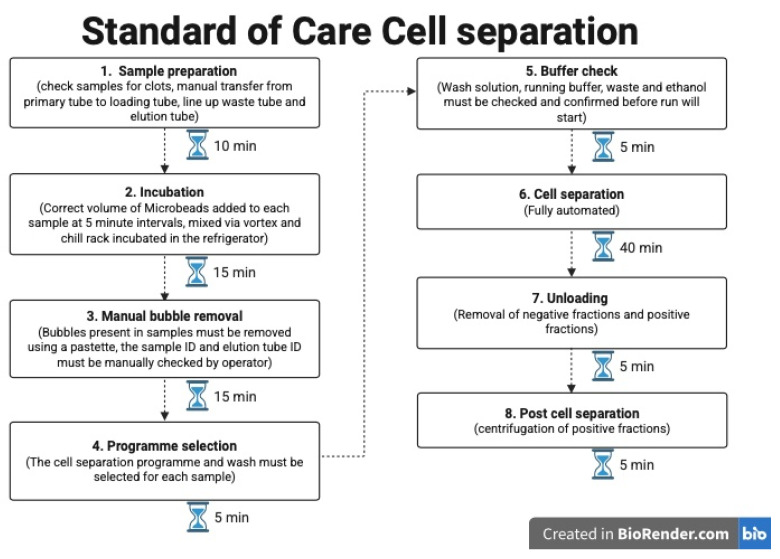
Flowchart showing the steps involved when loading the AutoMACS Pro, which is currently used as our standard-of-care approach (created in BioRender version 201). Time required for each step indicated and for performing cell separations on 5 samples.

**Figure 5 ijms-26-06747-f005:**
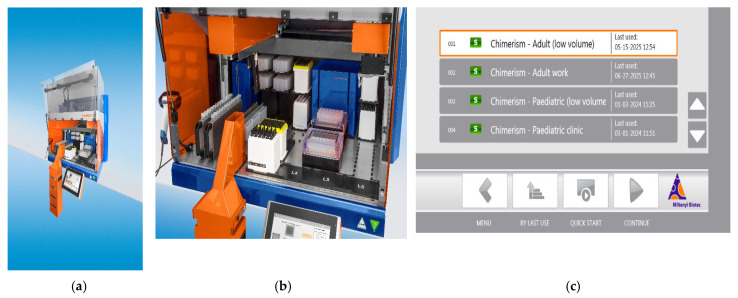
(**a**–**c**) Images of the MultiMACS X supplied by Miltenyi Biotec (**a**,**b**). Menu example displaying available custom programmes (**c**). The volume required for cell separation is dependent on the selected programme, all of which were designed to suit our laboratory sample requirements.

**Table 1 ijms-26-06747-t001:** Comparison of the purities obtained when separating CD3+ T cells, CD15+ granulocytes, and CD19+ B cells, in parallel using standard-of-care methods and the MMX for 20 samples.

	CD3+ T Cells		CD15+ Granulocytes		CD19+ B Cells	
Sample ID	Standard of Care Purity (%)	MMX Purity (%)	Standard of Care Purity (%)	MMX Purity (%)	Standard of Care Purity (%)	MMX Purity (%)
1	96	98	100	95	84	99
2	99	99	98	99	94	99
3	97	96	99	98	92	98
4	81	89	99	99	88	100
5	99	99	99	100	77	82
6	100	99	99	100	71	77
7	100	91	100	100	87	73
8	78	78	100	100	82	87
9	99	63	99	100	83	67
10	98	98	99	97	69	69
11	83	99	97	100	100	100
12	91	97	99	97	71	65
13	76	99	99	100	84	75
14	91	99	100	99	81	81
15	98	98	99	98	73	87
16	86	92	100	100	91	91
17	100	98	97	100	85	98
18	81	85	98	99	93	94
19	62	58	99	99	89	90
20	78	87	99	100	93	94

**Table 2 ijms-26-06747-t002:** Datasets for T cell, B cell, and granulocyte recovery following cell sorting using the MMX.

	Cell Fractionation Type		
	CD3+ T Cells	CD15+ Granulocytes	CD19+ B Cells
Median Cell Recovery (%)	58	68	78
Recovery Range (%)	37–81	30–99	48–99
>50% Cell Recovery	73	82	91

## Data Availability

The original contributions presented in this study are included in the article. Further inquiries can be directed to the corresponding author(s).

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
