# Peer review of "Validation of a New High-Throughput Cell Separation Method for Downstream Molecular Applications"

_ijms, 2025, doi:10.3390/ijms26146747_

Round 1

Reviewer 1 Report

Comments and Suggestions for Authors

Questions and Recommendations

The presented study is focused on the verifying a new method of cell separation on a large scale in clinical laboratories. The paper is very interesting as the new method could facilitate the separation procedure of high number of samples with sufficient purity for downstream applications that may save the time and reduce the handling errors. Anyway, the data presented in the paper are not sufficient for full research article type. I would like to recommend changing the article type to “Communication”, which is more suitable for new methods or technology.

Moreover, to improve the quality of provided manuscript, some important issue and discrepancies in results and methodology should be solved.  In conclusion, a major revision is required before accepting this paper for publication in International Journal of Molecular Sciences.

Below are listed some minor as well as major comments that should be corrected and clarified before the final decision:

Introduction:

Lines 33-34: The word “MiniMACS” is divided incorrectly.

Results:

Lines 86-87: The authors chose several key metrics to determine the performance of MMX method including (1) the purity of the sorted cells, (2) cell recovery, and (3) time required per sample for sorting. However, according to general experiences the microbeads are sticky to dead cells and debris. Therefore, cell viability in sorted samples should be also a very important metric. Even more, when sorted cells are used for downstream applications. Higher number of dead cells can negatively affect the results of such analyses. Thus, I recommend including of cell viability data of sorted samples within the presented results.

Figure 1a: The example dot plot (upper left) showed more than 90% of cell viability in all figures (1b and 1c) for MMX method. How was the cell viability in samples sorted with autoMACS Pro separator?

Upper right dot plot showed only 30% of leukocytes (CD45+ cells) in samples sorted for CD3 (T cells), or about 50% in samples sorted for CD15 (granulocytes) or CD19 (B cells). This is really low number of specific leukocytes. What are the other cells in the samples? Can the other cell subsets affect the objectivity of downstream applications?

Lower dot plots showed the contamination of specifically sorted cells with other cell subsets. Why are the CD3+ T cells contaminated with monocytes or B-cells? The same question can be asked for the contamination in other sorting cells (CD15+ or CD19+)? What is the real efficiency of such sorting?

In figure 1a legend, authors stated that CD3+ T cell purity in the example sample was 98.8%. However, this T cell purity belongs to the viable leukocyte subset which represents only 30% of cells within the CD3+ sorted samples. The same goes for CD15+ and CD19+ sorted cells where leukocytes represent only 50% from the whole sorted cell population. Therefore, I must ask what was the real yield or purity of specific cell subsets (CD3+, CD19+ or CD15+) in the sorted patient blood samples?

Figure 1c: Upper middle dot plot showed more than 60% of debris after gating of viable cells (upper left). Could you please explain this discrepancy? Moreover, an axe “x” is missing in all upper dot plots.

Line 107: “The full head-to-head comparison of T cell, B cell, and granulocyte purities between standard of care and MMX sorting is shown in Tables 1-3.“ However, Table 2 showed recovery and not purity data, and Table 3 is missing!!!

Line 122: “Figure 2. shows the linearity between the standard of care and MMX sorted cells.” This is text or Figure legend?

Line 125: “for (A) CD3+ T cells, (B) CD15+ Granulocytes, and (C) CD19+ B cells. Figure 2D shows…” Figures are not marked by A, B, C or D letters!!!

Line 174: The hand symbol is missing in Figure 3.

Line 176: “Overall, the time calculated per cell sort using the MMX was 5 minutes,” It is not clear for me from Figure 3, how did you calculate this time?

Discussion:

There are no references in Discussion!

The authors did not compare their findings with other published studies!

Materials and Methods:

Line 216: Please include the number of patients and their overall health status or diagnosis if available.

Line 217: Image of MMX instrument should be interesting for the readers.

Lines 218-219: Can you specify a custom-made programme?

Line 219, 220, 221 or elsewhere in methodology: The number and units should be separated by space (e.g. 2.7 ml; 0.9 ml or 7 ml).

Line 247: “using the autoMACS Pro, using the previously described protocol.” Did you use the same sample volumes as for MMX? Please specify autoMACS Pro programme used for sorting.

Line 250: Please specify g-force instead of rpm.

Line 255: Please specify the calculation of cell recovery by words or by an example calculation.

Author Response

Dear Reviewer 1,

We thank you for your time spent reviewing our submitted manuscript. We have made all the changes you suggested and now think the manuscript is much improved. We provide a comment by comment response below.

The presented study is focused on the verifying a new method of cell separation on a large scale in clinical laboratories. The paper is very interesting as the new method could facilitate the separation procedure of high number of samples with sufficient purity for downstream applications that may save the time and reduce the handling errors. Anyway, the data presented in the paper are not sufficient for full research article type. I would like to recommend changing the article type to “Communication”, which is more suitable for new methods or technology.

We are happy with this being categorized as a Communication

Moreover, to improve the quality of provided manuscript, some important issue and discrepancies in results and methodology should be solved.  In conclusion, a major revision is required before accepting this paper for publication in International Journal of Molecular Sciences.

Below are listed some minor as well as major comments that should be corrected and clarified before the final decision:

Introduction:

Lines 33-34: The word “MiniMACS” is divided incorrectly.

We have amended this.

Results:

Lines 86-87: The authors chose several key metrics to determine the performance of MMX method including (1) the purity of the sorted cells, (2) cell recovery, and (3) time required per sample for sorting. However, according to general experiences the microbeads are sticky to dead cells and debris. Therefore, cell viability in sorted samples should be also a very important metric. Even more, when sorted cells are used for downstream applications. Higher number of dead cells can negatively affect the results of such analyses. Thus, I recommend including of cell viability data of sorted samples within the presented results.

We agree. Although our intended downstream use is molecular testing it might be interesting for the reader to have a summary of viability. Thus, we have now included the cell viability data in the results section (section 2.1.1, lines 100-104).

Figure 1a: The example dot plot (upper left) showed more than 90% of cell viability in all figures (1b and 1c) for MMX method. How was the cell viability in samples sorted with autoMACS Pro separator?

We have now included the cell viability data in the results section (section 2.1.1, lines 100-104). Cell viability was established using 7AAD staining on the CD45+ cell population. This is now included in the figure 1a-c legend (page 5, lines 132-142) and also in the Methods section (page 12, line 347).

Upper right dot plot showed only 30% of leukocytes (CD45+ cells) in samples sorted for CD3 (T cells), or about 50% in samples sorted for CD15 (granulocytes) or CD19 (B cells). This is really low number of specific leukocytes. What are the other cells in the samples? Can the other cell subsets affect the objectivity of downstream applications?

The method requires no pre-processing of primary blood samples. This means there is substantial numbers of non-nucleated red cells and platelets in the sorted samples. We have added a line into the methods stating this (page 10, lines237-238). The methods section has also been updated to explain in greater detail that the CD45 negative gate represents red blood cells, platelets and debris which are cells without DNA, therefore not impacting downstream molecular testing (page 12, lines 347-349).

Lower dot plots showed the contamination of specifically sorted cells with other cell subsets. Why are the CD3+ T cells contaminated with monocytes or B-cells? The same question can be asked for the contamination in other sorting cells (CD15+ or CD19+)? What is the real efficiency of such sorting?

We agree that the plots do not explain data clearly. We have amended figures 1a accordingly. In Figure 1b we have added to the figure legend to explain that the monocyte and lymphocyte gate in the lower middle plot is derived from the gate in the CD15 negative population shown in the lower left plot. In figure1c we have changed the figure legend to explain that the lower middle plot is derived from the “non B-Cell”  gate in the lower left plot. The lower right plot is derived from the “non-monocyte/non-B-Cell gate”. We have added in roman numeral numbering for each individual plot to make it easier for the reader. We have also included arrows showing which population each subsequent gate is derived from.

In figure 1a legend, authors stated that CD3+ T cell purity in the example sample was 98.8%. However, this T cell purity belongs to the viable leukocyte subset which represents only 30% of cells within the CD3+ sorted samples. The same goes for CD15+ and CD19+ sorted cells where leukocytes represent only 50% from the whole sorted cell population. Therefore, I must ask what was the real yield or purity of specific cell subsets (CD3+, CD19+ or CD15+) in the sorted patient blood samples?

As previously described, in response to a previous comment we have updated the methods to explain that primary EDTA blood was labelled, without red cell lysing or ficol prep. The CD45 negative gate represents red blood cells, platelets and debris which are cells without DNA, therefore not impacting downstream molecular testing. This means the purity we present is the true purity once non-nucleated cells and debris are removed.

Figure 1c: Upper middle dot plot showed more than 60% of debris after gating of viable cells (upper left). Could you please explain this discrepancy? Moreover, an axe “x” is missing in all upper dot plots.

As previously described, in response to a previous comment we have updated the methods to explain that primary EDTA blood was labelled, without red cell lysing or ficol prep. The CD45 negative gate represents red blood cells, platelets and debris which are cells without DNA, therefore not impacting downstream molecular testing. This means the purity we present is the true purity once non-nucleated cells and debris are removed.

We have also corrected the axis.

Line 107: “The full head-to-head comparison of T cell, B cell, and granulocyte purities between standard of care and MMX sorting is shown in Tables 1-3.“ However, Table 2 showed recovery and not purity data, and Table 3 is missing!!!

We apologise, this was an error during manuscript formatting. All data was inputted into table 1 rather than separated based on cell population. This has been corrected.

Line 122: “Figure 2. shows the linearity between the standard of care and MMX sorted cells.” This is text or Figure legend?

We are sorry, we made a formatting error and some text was duplicated. The figure legend underneath left and the text above have now been removed.

Line 125: “for (A) CD3+ T cells, (B) CD15+ Granulocytes, and (C) CD19+ B cells. Figure 2D shows…” Figures are not marked by A, B, C or D letters!!!

We have corrected this and have now added the missing letters.

Line 174: The hand symbol is missing in Figure 3.

The hand symbol was removed prior to submission as we felt it did not add anything to the figure. We had forgotten to correct this in the figure legend. We have now updated the text in the figure legend to reflect this.

Line 176: “Overall, the time calculated per cell sort using the MMX was 5 minutes,” It is not clear for me from Figure 3, how did you calculate this time?

We have updated this time to 7 minutes based on an assumption that the full 24 sample capacity of the MMX is utilised in each run. Thus, we took the time required for each step and divided this by 24 (total number of samples). This works out at 6.33333 minutes per sample which we have rounded up. This is now included in the Results section (page 9, lines 218-222).

Discussion:

There are no references in Discussion!

The authors did not compare their findings with other published studies!

We have now included several papers into the discussion to justify some of our comments.

Materials and Methods:

Line 216: Please include the number of patients and their overall health status or diagnosis if available.

We have included a summary of the clinical status of the 20 patient samples used in this study (Methods section, page 10, lines 260-264).

Line 217: Image of MMX instrument should be interesting for the readers.

We agree and have now added a new figure showing the MMX, and the control panel with our bespoke program options. This is now Figure 5 (Methods section, page 11).

Lines 218-219: Can you specify a custom-made programme?

We have now included an image (Figure 5). We have also included this as a line in the Methods section (page 10, line 266).

Line 219, 220, 221 or elsewhere in methodology: The number and units should be separated by space (e.g. 2.7 ml; 0.9 ml or 7 ml).

We have now corrected throughout.

Line 247: “using the autoMACS Pro, using the previously described protocol.” Did you use the same sample volumes as for MMX? Please specify autoMACS Pro programme used for sorting.

The same volumes were used, text has been updated to state this. The programme used on the autoMACS Pro has been added.

Line 250: Please specify g-force instead of rpm.

We have now corrected to g-force.

Line 255: Please specify the calculation of cell recovery by words or by an example calculation.

We have now added this into the text (Methods section, page 11, lines 333-341).

Reviewer 2 Report

Comments and Suggestions for Authors

I read the manuscript by Shillingford et al in which authors introduce a new platform for larger scale cell sorting. The manuscript carries novelty and will be of interest to the readership given the growing application of technologies performing downstream bulk and single cell analysis which requires cell sorting beforehand. However, revisions should be undertaken before the consideration of acceptance for publication. Please, see my comments below, including major and minor ones:

-Introduction: At the end of the introduction, give a context of the analysis performed, for example, origin of samples e.g. human whole blood/PBMCs and the main indication/rationale of cell sorting e.g. chimerism post transplant. The reader has to go to the “Materials and Methods” section to understand the analysis context which should be clearly stated from the beginning.

-Section 2.1.1 Purity of sorted cells: Add some data on the main text. Data info is only provided in the figures.

-Figure 1: Add the a,b,c letters to each subfigure and include only one legend for the entire figure 1. I would also suggest adding arrows from each plot to the next one within a subfigure to showcase the steps of the gating strategy. Plots should also be aligned within each subfigure, currently they are not aligned.

-Figure 1.b: The legend states that CD19 was used as a marker to define B cells within CD15 cells. However, in the X axis of the last plot, CD20 is shown as a lineage B cell marker. Please, correct and use the same marker, either CD19 or CD20 to gate B cells in all figures. Currently CD19 is presented in Figure 1a and CD20 in Figure 1b and 1c.

-Figure 1.c: The panel markers on X axis are not shown at the first 3 plots. In addition, the figure legend mentions “CD19+ B cells” while CD20 is presented in the plot. Please, correct.

-Line 107: Correct Table 1-3 to Table 1.

-Table 1: Do the authors have an explanation why there was a significant difference in the purity between SOC and MMX for CD3+ T cells and CD19+ B cells in sample 9?

-Line 119: Correct this sentence to include the difference in purity seen in sample 9 for CD19+ B cells.

-Line 122 should move before the images.

-Figure 2: Please add the letters A, B, C and D to the relevant graph within the figure.

-Section 2.1.3: Please, present interquartile range (IQR) as a measure of statistical dispersion for each comparison.

-Section 2.2 and Table 2: Unclear whether the presented data refer to MMX method. I suggest authors present data for both SOC and MMX method and perform statistical tests for comparisons.

-Figure 3: Add explanation of the abbreviations DWPs and ID’s at the legend of the figure, even if they are well recognised on what they stand for. In addition, there are no hand symbols in the image to highlight the manual handling. Please, correct. It would be interesting to present a similar flowchart of the equivalent times needed for each step at the SOC method to directly compare with the MMX method.  

-Line 193-195: Add citations if available.

-Discussion section: Probably cell viability is not necessary for chimerism analysis prior and post cell sorting. However, given that this method could be implemented for research purposes or other clinical fields, add comments for any potential effect this new method may have on cell viability or alternation on cell phenotype eg T cell subsets. Please, also add any limitations of the method such as non-specific binding or activation artifacts in another type of clinical application.

-Section 4.1. MultiMACS X Cell Fractionation and 4.2. Standard of Care Cell Fractionation: Further details should be provided. Please, add the manufacturer and catalog number of MACS. Specify what antibodies or ligands were conjugated to the beads. Specify labeling conditions such as incubation time, temperature and bead-to-cell ratio. Was any blocking reagent used to minimize non-specific binding? Was RBC lysis included as a step for sample preparation?

-Lines 264-270: Can the authors explain why they have not used the same marker, CD19 or CD20, for B cells? I have also commented above on this matter regarding Figure 1.

-4.4 Statistical Analyses: Add the cut off of p value that was used to define statistical significance in comparisons. In addition, the authors used the Wilcoxon test for statistical comparisons. However, this test is used to compare paired samples e.g. before and after an intervention. The authors have not undertaken this type of analysis for purity data. Please, correct the analysis using the non-parametric Mann-Whitney test which seems more appropriate when comparing independent samples.

-Add a Conclusion section as a final section and highlight there key findings presented in the manuscript and significance of them.

Author Response

Dear Reviewer 2,

We thank you for your time spent reviewing our submitted manuscript. We have made all the changes you suggested and now think the manuscript is much improved. We provide a comment by comment response below.

-Introduction: At the end of the introduction, give a context of the analysis performed, for example, origin of samples e.g. human whole blood/PBMCs and the main indication/rationale of cell sorting e.g. chimerism post transplant. The reader has to go to the “Materials and Methods” section to understand the analysis context which should be clearly stated from the beginning.

We have added an extra statement in the introduction to contextualise that testing was performed on human whole blood samples for downstream molecular testing - such as cell lineage chimerism analysis post-transplant (page 2, Introduction, lines 81-85).

We agree and the clinical status of samples is now included in the Materials and Methods section (page 10, lines 260-264).

-Section 2.1.1 Purity of sorted cells: Add some data on the main text. Data info is only provided in the figures.

We have now added some text to expand on purity data as requested (section 2.1.1, page 3, lines 97-104). We have included median and mean of the data.

-Figure 1: Add the a,b,c letters to each subfigure and include only one legend for the entire figure 1. I would also suggest adding arrows from each plot to the next one within a subfigure to showcase the steps of the gating strategy. Plots should also be aligned within each subfigure, currently they are not aligned.

We have now labelled the plots as requested (a, b, and c) and to make it easier for the reader we have also added subfigure numbering (I, ii, iii, iv, v, and vi) and referred to these in the figure legend. We have used arrows to show which populations each gate is derived from to make it easier to follow. We have made one legend as suggested.

-Figure 1.b: The legend states that CD19 was used as a marker to define B cells within CD15 cells. However, in the X axis of the last plot, CD20 is shown as a lineage B cell marker. Please, correct and use the same marker, either CD19 or CD20 to gate B cells in all figures. Currently CD19 is presented in Figure 1a and CD20 in Figure 1b and 1c.

We apologise for the typo and have corrected to CD20 in the legend. We were unable to use CD19 for of purity assessment of sorted CD19+ B cells because of the blocking of the CD19 epitope specific to these microbeads. We had to follow manufacturer recommendations and use CD20 as a marker for purity assessment with the sorted B cells. We have added a comment in the results section (section 2.1.1, page 3, lines 94-97).

-Figure 1.c: The panel markers on X axis are not shown at the first 3 plots. In addition, the figure legend mentions “CD19+ B cells” while CD20 is presented in the plot. Please, correct.

We apologise, this was lost during manuscript formatting. We have now corrected the text to state B-cell purity.

-Line 107: Correct Table 1-3 to Table 1.

We apologise, this was an error during manuscript formatting. All data was inputted into table 1 rather than separated based on cell population. This has been corrected.

-Table 1: Do the authors have an explanation why there was a significant difference in the purity between SOC and MMX for CD3+ T cells and CD19+ B cells in sample 9?

Sample 9 was collected and sorted at identical times by both methods. However, there was a delay of 24 hours between SoC purity assessment and MMX purity assessment for the T and B cells due to workload commitments. We have now added a comment to explain there was a delay in sample purity processing which may have contributed to the low purities in the MMX sorted T and B cells (page 6, lines 153-156)

-Line 119: Correct this sentence to include the difference in purity seen in sample 9 for CD19+ B cells.

See comment above regarding Sample 9.

-Line 122 should move before the images.

We have now moved the table caption.

-Figure 2: Please add the letters A, B, C and D to the relevant graph within the figure.

We have added the missing letters.

-Section 2.1.3: Please, present interquartile range (IQR) as a measure of statistical dispersion for each comparison.

We have calculated and added the IQR for each dataset (page 7, 187-189).

-Section 2.2 and Table 2: Unclear whether the presented data refer to MMX method. I suggest authors present data for both SOC and MMX method and perform statistical tests for comparisons.

We have updated Table 2 to show the recovery data generated after MMX cell sorting. Due to limited sample availability, it was not possible to generate cell recovery data for SoC method. However, this has been documented previously as shown by Willasch et al.(ref 19), and we have included this in the discussion (page 9, lines 237-239).

-Figure 3: Add explanation of the abbreviations DWPs and ID’s at the legend of the figure, even if they are well recognised on what they stand for. In addition, there are no hand symbols in the image to highlight the manual handling. Please, correct. It would be interesting to present a similar flowchart of the equivalent times needed for each step at the SOC method to directly compare with the MMX method.  

Comment added to identify abbreviations in the figure legend. The hand symbol was removed prior to submission as we felt it did not add anything to the figure. We had forgotten to correct this in the figure legend. We have now updated the text in the figure legend to reflect this.

Additional flowchart added for SoC as requested (figure 4).

-Line 193-195: Add citations if available.

These are our calculated hands-on times and we have extrapolated our calculations to explain this (page 9, lines 219-223).

-Discussion section: Probably cell viability is not necessary for chimerism analysis prior and post cell sorting. However, given that this method could be implemented for research purposes or other clinical fields, add comments for any potential effect this new method may have on cell viability or alternation on cell phenotype eg T cell subsets. Please, also add any limitations of the method such as non-specific binding or activation artifacts in another type of clinical application.

We have updated the discussion to state that although cell viability is not vital for downstream molecular testing such as chimerism analysis it has been noted that both cell recovery and the viability of cells following MACS are high, therefore this may be of interest in research settings. We have also included a reference highlighting the impact of MicroBead sorting on cell function, structure and activity.  (page 10, lines 246-250).

-Section 4.1. MultiMACS X Cell Fractionation and 4.2. Standard of Care Cell Fractionation: Further details should be provided. Please, add the manufacturer and catalog number of MACS. Specify what antibodies or ligands were conjugated to the beads. Specify labeling conditions such as incubation time, temperature and bead-to-cell ratio. Was any blocking reagent used to minimize non-specific binding? Was RBC lysis included as a step for sample preparation?

The methods for both the MultiMACS X and Standard of Care Cell Fractionation have been expanded to include further detail regarding the MicroBeads and the conditions required for use (pages 10,11 lines 289 – 296 and page 11, lines 319-322). No additional blocking reagents or lysis steps used for either method.

-Lines 264-270: Can the authors explain why they have not used the same marker, CD19 or CD20, for B cells? I have also commented above on this matter regarding Figure 1.

We have addressed this in the results section. The manufacturer has confirmed that because of the binding of the MicroBeads to the CD19 epitope it is not possible to assess B cell purity using another CD19 marker. Therefore, CD20 was used as a B cell marker for purity assessment as per manufacturers kit insert (page 3, 95-97).

-4.4 Statistical Analyses: Add the cut off of p value that was used to define statistical significance in comparisons. In addition, the authors used the Wilcoxon test for statistical comparisons. However, this test is used to compare paired samples e.g. before and after an intervention. The authors have not undertaken this type of analysis for purity data. Please, correct the analysis using the non-parametric Mann-Whitney test which seems more appropriate when comparing independent samples.

4.4 The statistical analyses have been updated to include the cut off p value for statistical significance (page 12, lines 363-365).

The Mann-Whitney test has now been utilised and included in 2.1.3 Statistical analysis of purity data (page 7, lines 171 – 176).

-Add a Conclusion section as a final section and highlight there key findings presented in the manuscript and significance of them.

Conclusion added to summarise key findings presented in the manuscript and their significance.

Round 2

Reviewer 2 Report

Comments and Suggestions for Authors

Thank you for addressing all comments. I would suggest some further minor corrections.

-Section 2.1.1 Purity of sorted cells: Add some data on the main text. Data info is

only provided in the figures.

We have now added some text to expand on purity data as requested (section 2.1.1,

page 3, lines 97-104). We have included median and mean of the data.

Usually either median or mean data are presented. Given that variables are not normally distributed, please present only median and remove mean data to avoid duplication.

-Section 2.1.3: Please, present interquartile range (IQR) as a measure of

statistical dispersion for each comparison.

We have calculated and added the IQR for each dataset (page 7, 187-189).

Thank you for performing this analysis which is always valuable in understanding data dispersion. Typically, data is presented in the form of median (25th IQR, 75th IQR). Can you please present the data in the above form for each comparison? 

-4.4 Statistical Analyses: Add the cut off of p value that was used to define

statistical significance in comparisons. In addition, the authors used the

Wilcoxon test for statistical comparisons. However, this test is used to compare

paired samples e.g. before and after an intervention. The authors have not

undertaken this type of analysis for purity data. Please, correct the analysis using

the non-parametric Mann-Whitney test which seems more appropriate when

comparing independent samples.

4.4 The statistical analyses have been updated to include the cut off p value for

statistical significance (page 12, lines 363-365).

The Mann-Whitney test has now been utilised and included in 2.1.3 Statistical

analysis of purity data (page 7, lines 171 – 176).

  I can see that results after applying the Wilcoxon test are still included in the manuscript. Wilcoxon is not a suitable statistical test given that independent and not paired samples are compared. If authors insist to leave this analysis, can they please provide a rationale for this? Otherwise, this part should be removed. In addition, please add the statistical Mann-Whitney test in the “Statistical Analysis” subsection of the “Materials and Methods” 4 section.   In this same section, the authors have performed a correlation coefficient analysis using the Pearson test. To start with, given that data are not normally distributed, the Spearman test should be used instead of the Pearson. However, this type of analysis does not make sense in the context of correlating SoC purity data with MMX relevant data. Usually, such a correlation is investigated to find a relationship or causality between variables. I would suggest to remove this analysis from the manuscript.

Author Response

Thank you for addressing all comments. I would suggest some further minor corrections.

-Section 2.1.1 Purity of sorted cells: Add some data on the main text. Data info is

only provided in the figures.

We have now added some text to expand on purity data as requested (section 2.1.1,

page 3, lines 97-104). We have included median and mean of the data.

Usually either median or mean data are presented. Given that variables are not normally distributed, please present only median and remove mean data to avoid duplication.

 We agree and have now removed mean.

-Section 2.1.3: Please, present interquartile range (IQR) as a measure of

statistical dispersion for each comparison.

We have calculated and added the IQR for each dataset (page 7, 187-189).

Thank you for performing this analysis which is always valuable in understanding data dispersion. Typically, data is presented in the form of median (25th IQR, 75th IQR). Can you please present the data in the above form for each comparison? 

We have now formatted this correctly.

-4.4 Statistical Analyses: Add the cut off of p value that was used to define

statistical significance in comparisons. In addition, the authors used the

Wilcoxon test for statistical comparisons. However, this test is used to compare

paired samples e.g. before and after an intervention. The authors have not

undertaken this type of analysis for purity data. Please, correct the analysis using

the non-parametric Mann-Whitney test which seems more appropriate when

comparing independent samples.

4.4 The statistical analyses have been updated to include the cut off p value for

statistical significance (page 12, lines 363-365).

The Mann-Whitney test has now been utilised and included in 2.1.3 Statistical

analysis of purity data (page 7, lines 171 – 176).

  I can see that results after applying the Wilcoxon test are still included in the manuscript. Wilcoxon is not a suitable statistical test given that independent and not paired samples are compared. If authors insist to leave this analysis, can they please provide a rationale for this? Otherwise, this part should be removed. In addition, please add the statistical Mann-Whitney test in the “Statistical Analysis” subsection of the “Materials and Methods” 4 section.   In this same section, the authors have performed a correlation coefficient analysis using the Pearson test. To start with, given that data are not normally distributed, the Spearman test should be used instead of the Pearson. However, this type of analysis does not make sense in the context of correlating SoC purity data with MMX relevant data. Usually, such a correlation is investigated to find a relationship or causality between variables. I would suggest to remove this analysis from the manuscript.

Thank you for the guidance. We have now removed the Wilcoxon and Pearson analyses. We have added in the Mann-Whitney U Test to the methods section.